# Prevalence of Human Papillomavirus (HPV) among Females in the General Population of the Split and Dalmatia County and Its Association with Genital Microbiota and Infections: A Prospective Study

**DOI:** 10.3390/v15020443

**Published:** 2023-02-05

**Authors:** Vanja Kaliterna, Petar Kaliterna, Lidija Pejkovic, Robert Vulic, Linda Zanchi, Karmen Cerskov

**Affiliations:** 1Teaching Institute for Public Health of Split and Dalmatia County, 21000 Split, Croatia; 2University Department of Health Studies, University of Split, 21000 Split, Croatia; 3School of Medicine, University of Split, 21000 Split, Croatia; 4Gynecological Practice, 21210 Solin, Croatia; 5Gynecological Practice, 21000 Split, Croatia; 6Gynecological Practice, 21212 Kastel Sucurac, Croatia

**Keywords:** HPV prevalence, Croatia, cervicovaginal microbiota, *Ureaplasma*, *Chlamydia*, *Gardnerela*, cervical dysplasia

## Abstract

We conducted a prospective study with the aim of determining HPV prevalence and type distribution in the general female population of Southern Croatia (SDC), and to detect the presence of other microorganisms in the lower part of the female reproductive system and their possible influence on the frequency of HPV infection. Data were collected during routine check-up exams. All participants were examined by a gynecologist, and cervico-vaginal scrapings/swabs were collected, for cytological (Pap smear) and microbiological (for bacterial growth, genital mycoplasmas, chlamydia, and HPV) analysis. Informed consent was obtained from all participants with accompanying questionnaire. A total of 1050 asymptomatic women living in SDC participated in the study during a one-year period, and 107 of them (10.2%) had HR-HPV infection. We found that the presence of some bacteria (*Ureaplasma*, *Chlamydia*, and *Gardnerella*) in the lower part of the female genital system has a positive correlation with the frequency of HPV infection and, consequently, a possible influence on faster progression to cervical dysplasia caused by HPV. We consider that inclusion of screening for sexually transmitted infections as monitoring in women with HPV infection could help to find women at risk of cervical cancer progression.

## 1. Introduction

Cervical cancer (CC) is the fourth cause of cancer in women worldwide with 604,127 new cases (13.3/100,000) and the fourth leading cause of cancer death in women with 341,231 deaths (7.3/100,000) in 2020 [1]. In 2020, CC was ranked as the 9th most frequent cancer among women (58,169 new cases) and the 10th cause of cancer death in women (25,989 deaths) in Europe, but it is the 3rd most common female cancer in women aged 15 to 44 years in Europe [2]. In total, 85% of cases occur in developing countries because of absent or poor screening programs for CC prevention. Infection with Human Papillomavirus (HPV) is the main cause of cervical cancer, and it is detected in 99.7% of all cervical malignancies [3,4]. HPV is also one of the most frequent causative agents of sexually transmitted diseases (STDs) globally. It can cause various clinical conditions, from asymptomatic infections to benign and malignant diseases of the genital region. Most HPV infections are spontaneously cleared by the host immune response. At present, more than 200 types of HPV have been identified. More than 40 types attack the epithelium of the genital tract and are transmitted mainly through sexual intercourse. Types that are most commonly related to the development of cervical cancer, so-called oncogenic or high-risk types (HR HPV), are: 16, 18, 31, 33, 34, 35, 39, 45, 51, 52, 56, 58, 59, 66, 68, and 70. HPV-16 and HPV-18 are associated with 70% of cervical cancer worldwide [3,4]. The prevalence and distribution of the different HPV types in cervical samples vary around the world. It is estimated that the prevalence of HPV infection is between 9% and 13% of the world population [5]. Genital HPV infection is rarely reported (there is no legal obligation). According to the Croatian National Cancer Registry and the latest data for Croatia, in 2020, there were about 276 new cases of cervical cancer (incidence rate 13.3/100,000) and about 122 deaths (mortality rate 6.1/100,000 women) [6].

The presence of HPV is necessary for CC developing but not sufficient cause of cervical cancer. The development of HPV-induced cancer is associated with some co-factors, such as other sexually transmitted infections (STIs) and even the presence of a high diversity in the cervico-vaginal microbiota (CVM) [1,7].

There are only a few studies about the prevalence and distribution of HPV types in Croatia, based on laboratory test results, related to cervical samples of selected women, referred by their gynecologists to laboratory for HPV DNA testing, with either a normal or abnormal Pap smear [8,9].

The aims of the present study were to determine the prevalence of HPV and HPV distribution of HPV-16 and -18 and 12 other HR HPV genotypes in the general female population of Southern Croatia (Split and Dalmatia County, SDC), to detect the presence of other microorganisms in the lower part of the female reproductive system (either as part of the normal CVM or as causative agents of genital infections) and their possible influence on the frequency of HPV infection, as well as to determine the association between HPV infection and abnormal Pap test results.

## 2. Materials and Methods

We performed a prospective study to determine HPV prevalence and type distribution of outpatient women in the SDC. Data were collected during routine check-up exams by a group of gynecologists across the county, from four gynecological practices of different parts of the county, that were unaffiliated with the study. Selection bias was eliminated because the gynecologists sent samples of all women (with previous sexual experience within their lifetime) who came for regular gynecological examination on specific predetermined days over a period of one year. Exclusion criteria were pregnancy, lower genital tract dysplasia, or malignancy and if they suffered from more severe diseases (e.g., carcinoma and immunodeficiency).

The study involved a total of 1050 women living in SDC, during a one-year period. All participants underwent an examination performed by a gynecologist, and cervico-vaginal scrapings/swabs were collected for cytological (Pap smear) and microbiological (for bacterial growth, genital mycoplasmas, chlamydia, and HPV) analysis [10,11]. A structured questionnaire was obtained from all participants. The first section asked for basic sociodemographic details (age, marital status). The second part was focused on inquiring about the participant’s personal gynecological history (number of pregnancies, use of contraceptives, and number of sexual partners in the last 6 months). All procedures were explained to the respondents and they signed an informed consent to agree that their results can be processed anonymously for the purpose of this work.

Bacterial cultivations were performed from the material obtained from cervical swabs using standard laboratory procedures. Normal healthy CVM was defined as vaginal homeostasis with *Lactobacillus* spp. as dominant bacterium. On the other hand, disruption of vaginal homeostasis we marked as CVM dysbiosis, which meant some other bacterial growth in pure culture or in the predominant number (mainly bacteria associated with bacterial vaginosis (BV)). Detection of genital *Mycoplasma hominis* and *Ureaplasma urealyticum* were performed using the commercially available Mycoview kit (*Zeakon* diagnostics, France) with a cut-off value of 10^4^ CCU/mL. *Chlamydia trachomatis* (CT) and HR-HPV were detected in cervical exfoliated cells using a real-time FDA approved PCR assay (*Roche Cobas 4800 HPV Test*, Roche Molecular Systems, Pleasanton, USA) based on concurrent individual genotyping for HPV-16 and HPV-18 and the pooled detection of 12 other HR-HPV types. The cytology was reported using the Bethesda system, either as atypical squamous cells of undetermined significance (ASCUS), low grade squamous intraepithelial lesion (LSIL), or high grade squamous intraepithelial lesion (HSIL) [12].

The study has been approved by the Ethics Committee of the Teaching Institute for Public Health of Split and Dalmatia County, TIPH SDC, within which the work was performed (Class: 500-01/15-01/6, No: 2181-103-01-15-1). All parts of the study were performed in accordance with the Declaration of Helsinki.

### Statistical Analysis

In this study, all variables were qualitative. The age variable was defined for two age groups, namely for women aged 29 years or less and for women aged 30 years or more (as nominal). The obtained results were stored in the MS Excel database. Structural pie charts and tables were used to display the selected variables. To display the structure according to the modalities of the selected variables, in addition to nominal values, relative numbers in percentages are also used. The Kolmogorov–Smirnov test was used to check the data distribution. For the statistical processing of qualitative variables that had a normal distribution, the Chi-square (χ^2^) test was performed. Statistical analysis was performed using the statistical package SPSS 17.0 (SSPS Inc., Chicago, IL, USA). A *p*-value < 0.05 was considered statistically significant.

## 3. Results

In this study, in a one-year period, a total of 1050 asymptomatic women, aged 17–74 years, from the general population in SDC attending routine gynecological care, were evaluated (Table 1).

Out of the total number of tested women (*N* = 1050), 304 (29%) of them were ≤ 29 years old and 746 (71%) were ≥30 years old (Figure 1).

According to the data in Table 2, the most frequently isolated microorganisms were *Ureaplasma urealyticum* (Uu), in 317 out of 1050 women (30.2%). In our study, HR-HPV infection was proved in cervical samples of 107 (10.2%) women. Although the majority of tested women, i.e., 98% of them (1029), had a negative *Chlamydia trachomatis* (CT) test on regular check-up, the primary pathogenic bacterium CT was detected in 21 (2%) cases. Bacterial growth is a normal finding in a cervical swab, as it is often contaminated with CVM when the swab is taken. These bacteria of CVM are in balance in the mixed physiological flora of the lower parts of the genital system (mainly with *Lactobacillus* sp.) and do not cause infections. In our study, 869 (82.8%) women had such a normal flora, and we marked them in Table 2 as CVM dysbiosis negative. In 181 samples (17.2% of tested women), we detected dysbiosis (imbalance) in CVM due to growth of some bacteria (*Streptococcus agalactiae*, *Escherichia coli*, *Gardnerella vaginalis*) in the predominant number or in pure culture, half of which accounted for *Gardnerella vaginalis* (GV), in 90 samples.

Out of the total number of 1050 asymptomatic women on regular gynecological examination, 40 of them (3.8%) had an abnormal PAP result (including LSIL and HSIL), while 1010 of them (96.2%) had a normal cytological finding (Table 3).

No statistically significant dependence was found between the age of women and Pap test results. For women aged ≤29 years, 14 of 304 (4.6%) had an abnormal cytological result, while 26 (3.5%) of 746 women in the group aged ≥30 years had an abnormal result of the Pap smear (*p* = 0.3898).

Out of the total number of 1050 tested women, 107 of them (10.2%) had HR-HPV infection, while the rest of 943 (89.8%) women were negative for HR-HPV (Figure 2).

HR-HPV infection was statistically significant higher in women aged ≤29 years, in 19.4%, while in the group aged ≥30 years, 6.4% was HPV positive (*p* < 0.001) (Table 4).

We found that the presence of some bacteria (Uu, CT, and GV) in the lower part of the female reproductive system have positive correlation with frequency of HPV infection.

It is clear from the data in Table 5 that among those who had a positive cervical swab for *Ureaplasma*, there were a significantly higher proportion of women with a positive HPV test, i.e., in 19.2% samples, while among Uu-negative women, only 6.3% had an HPV infection (*p* < 0.001). Out of the 1029 women who were negative for CT test, only 9.7% was found to have a positive HPV test, while 33% of women who were CT positive had an active HPV infection at the same time. A statistically significant association between a positive result for CT and HPV infection was established (*p* < 0.001) (Table 5).

In our study, there were 181 women with detected CVM dysbiosis (Table 2 and Table 6). HPV was simultaneously confirmed in 12.7% cases with CVM dysbiosis, compared with 9.7% of women with normal genital microbiota Table 6. No statistically significant association between CVM dysbiosis and a positive HPV test result was found (*p* = 0.218598).

However, we noticed that the bacterium *Gardnerella vaginalis* (GV) was isolated in half of the women with CVM dysbiosis, i.e., in 49.7% (90 of 181 women with the disorder). We analyzed separately the connection between GV and HPV. According to the data in Table 5, it is clear that, among women who had a positive cervical swab for GV, there were a significantly higher proportion of those who had a simultaneous HPV infection, i.e., in 18.9% of samples. Among those who were GV negative, only 9.4% of women were HPV positive. This represents a statistically significant difference (*p* < 0.05).

In this study, we found a significant association between HPV infection and the abnormal cytology result. Among those who had a positive HPV test, 15% had an abnormal Pap test (LSIL or HSIL). In the group of HPV negative women, only 2.5% had an abnormal Pap smear. This represents a statistically significant difference (*p* < 0.001) (Table 7).

Out of the total number of asymptomatic participants from the SDC (*N* = 1050) who attended a regular gynecological visit, HPV-16 was detected in 38 (3.6%), HPV-18 in 7 (0.7%), while other HR-HPV genotypes were determined in 89 (8.5%) tested women. In this research, 107 (10.2%) women were HPV positive, and among positive samples, the following genotype distribution was detected: HPV-16 in 35.5% and HPV-18 in 6.5%, while the other HPV types were established in 83% (Table 8). Multiple HPV infections were found in 25% of positive specimens (27/107).

## 4. Discussion

The prevalence and distribution of the different HPV types in cervical samples vary around the world and is associated with the presence of cytological changes in the cervix [13]. At present, there are limited data regarding the prevalence and distribution of HPV types in Croatia. Before, we conducted a study for Southern Croatia (SDC) related to cervical samples of women with either normal or abnormal Pap smear, based on laboratory test results, and 35% of them were positive to HR HPV, with the most common type HPV 16 in 10.8% specimens [8]. On the other hand, the study from Northern Croatia (Zagreb County) has included cervical samples of women mostly with abnormal Pap smear. That was the reason for their higher HPV prevalence that ranges from 31.3% to 87% (depending on different grades of cytological results) with the most frequent type being HPV-16, in 19.8% samples [9].

The present study involved a total of 1050 women living in SDC, during a one-year period who attended routine gynecological visit, and 107 of them (10.2%) had proven HPV infection. Genotype HPV-16 was detected in 3.6% women and HPV-18 in 0.7%, while other HR-HPV genotypes were determined in 8.5% tested women. Our present results are different from those found in other studies in Croatia which looked at HPV prevalence, because they differ according to the class of observed female population. To our knowledge, this is the first population-based study in Croatia that analyzed the prevalence of HPV infection and genotype distribution in asymptomatic women from the general population.

The percentage of HPV-positive women in the present study is in agreement with the data published in the meta-analysis of a systematic literature review conducted by de Sanjose et al., who showed that, among women with regular cytology worldwide, 10.4% was HPV positive [13]. The majority of the women we tested had normal cytological smears (96.2%), while only 3.8% women had an abnormal Pap test, because the participants were from the general female population. Among 107 HPV positive women, there was a significantly higher proportion of women who had an abnormal Pap test (LSIL or HSIL), i.e., 15% of samples, while in the group of HPV negative women, only 2.5% had an abnormal Pap smear.

Despite being the most common STI and the causal agent of cervical cancer, it is still not clear why the majority of HPV infections resolve without consequences, while only a small proportion of them persist, causing intraepithelial lesions and cervical cancer [7]. Current literature confirmed the role of genital microbiota and STIs other than HPV as co-factors in cervical carcinogenesis caused by HPV, but this has not been fully elucidated and further research is needed [14]. Lv et al. considered that pathogens of the lower genital tract could cause chronic cervical infection, which would damage the mucosal barrier and immune protection, and thus, promote HR-HPV infection [15].

Of all the bacteria detected in genital swabs in the present study, after statistical analyses, we found that only *Ureaplasma urealyticum*, *Chlamydia trachomatis,* and *Gardnerella vaginalis* showed a significant association with HPV infection.

In the present study, 30.2% participants had positive cervical swab for *Ureaplasma urealyticum* (≥10^4^ CCU/mL). Among Uu-positive cases, there were a significantly higher proportion of women with a positive HPV test, i.e., in 19.2% of the samples, while among Uu-negative cases, only 6.3% of women had HPV infection. Our results are in agreement with previous research work from Greece, which showed that 25.4% of Uu-positive women simultaneously had an HPV infection. Some other authors have also found a relationship between Uu and HPV infection. Their results suggest that Uu infection increases the risk of HPV infection, which may have a synergistic effect on the development of HPV-related cervical dysplasia [14,16,17]. It has always been controversial whether *Ureaplasma* can be considered colonization or infection. A cut-off value of more than 10^4^ CCU/mL Uu in a sample is an additional criterion to distinguish colonization from infection, and Kim et al. also indicated that only a Uu with ≥ 10^4^ CCU/mL was significantly associated with HPV infection. They also found this relationship in asymptomatic women and, based on this, they recommended HR-HPV testing for women with Uu infection, regardless of symptom status because of a possible synergistic effect between *Ureaplasma* and HPV in cervical dysplasia caused by HPV [17].

Although the majority of individuals infected with *Chlamydia trachomatis* (CT) are asymptomatic, they can be carriers of CT infection, which can promote HR-HPV infection. In our study, only 2% women were CT positive and among them, 33% had HPV infection at the same time, while only 9.7% of HPV positive women was CT negative. Our results are in agreement with Lv et al., who revealed about four-fold higher risk of HR-HPV infection in CT-positive women compared with CT-negative women. They suggested that CT infection may damage the mucosal barrier and could increase cervical susceptibility to HPV and indicated the necessity of screening and treatment for CT in HR-HPV-positive women [15]. Franchini et al., in their review, concluded that the role of CT in CC is not yet fully understood, with some studies finding a correlation between cervical lesions and coinfection with CT and HPV, while others finding no association at all. They promoted annual CT screening worldwide because women are usually asymptomatic and can only be diagnosed and treated in time through screening [18].

The cervico-vaginal microbiota (CVM) has a symbiotic relationship with their host and plays a significant role in the health and disease of the female reproductive tract. It is the first line of defense against infections and appears to play a key role in the acquisition, persistence, and clearance of HPV infection. The heterogeneity of the cervico-vaginal microbiota can vary according to biological and environmental factors. Women with a specific vaginal microbiota composition may be more likely to acquire HPV or to show a faster progression to dysplasia [3,5,7,19,20]. A healthy CVM reflects the condition of vaginal homeostasis (a state of eubiosis) that is favored by the predominance of *Lactobacillus* spp. They produce lactic acid and hydrogen peroxide, and in that way, they maintain low pH and increase the viscosity of cervical mucus, and thus, protect the cervical epithelial barrier [5,21].

The researchers noted that the development of HPV-induced cancer is associated with a disruption of CVM homeostasis, which is involved in the control of viral persistence and is, therefore, a prognostic marker for disease. A state of vaginal dysbiosis or microbial imbalance indicates changes in the composition of CVM, with a decrease in mucus production, a decrease in the amount of *Lactobacillus* population, and an increase in CVM diversity, which is mainly composed of bacteria associated with BV, as well as often being associated with strict anaerobic species, including *Gardnerella vaginalis* (GV) [5,7,22]. Some authors found that the clearance of HPV was delayed among women who had BV, leading to an increasing progression of persistent infection to a precancerous lesion [15]. Abdalah et al. found that patients with established HR-HPV infection had a higher predominance of BV compared to those without infection [23]. The results of Usyk at al. suggested a novel association between the effect of GV and disruption of CVM homeostasis that can influence progression of HR-HPV infection to CC. They proposed a model in which GV causes an expansion of bacterial diversity, which acts as a risk factor for the progression of a HR-HPV infection into a CIN2+ lesion [19]. Gillet et al., in their meta-analysis, confirmed a positive association between BV and cervical pre-cancerous lesions [24].

In our present study, in 17.2% of tested women, CVM dysbiosis was detected, while 82.8% showed a normal genital microbiota. No statistically significant association between CVM imbalance and a positive HPV test result was found (HPV was detected in 12.7% of participants with CVM dysbiosis and in 9.7% of participants with normal genital microbiota). However, we noticed that the bacterium GV was isolated in half of the women with CVM dysbiosis, i.e., in 90 (49.7%). According to our results, it is clear that among 90 women who had a positive cervical swab for GV, a significantly higher proportion of those had an HPV infection at the same time, i.e., in 18.9%, while only 9.4% of women who were GV negative had an HPV infection. This result is in agreement with other authors who found an association between GV presence and HPV infection, as well as a highly positive correlation with precancerous progression. It is suggested that monitoring the presence of GV and the CVM diversity could help find women at risk for progression to cervical cancer [19].

In his paper, Mandeli emphasized the importance of simultaneous diagnosis of sexually transmitted infections in the genital tract as a key for determining unfavorable clinical outcomes of cervical cancer [13]. In the future, monitoring of the variability in the CVM diversity in the HPV screening program could be performed through new therapeutic strategies that determine the progression of the disease by changing CVM and activating a local immune response [19,20].

In our study, on more than 1000 participants of general population, we proved that CVM dysbiosis, mostly caused with *Gardnerella vaginalis*, can influence the susceptibility to HR-HPV infection (probably causing suppression of the local immune response, which can lead to greater susceptibility to HPV and, consequently, promote cervical dysplasia). Furthermore, we found that bacteria *Ureaplasma urealyticum* and *Chlamydia trachomatis* had a very high positive association with HR-HPV infection, and consequently, a possible synergistic effect in the development of cervical dysplasia caused by HPV. Therefore, we believe that it is necessary to raise the level of care for women’s health by introducing a screening program for both HPV and other STIs, which should be available to women worldwide. We did not perform testing for *Mycoplasma genitalium* (MG) at the time of conducting our study, but nowadays, MG is recognized as important as CT infection, and screening for MG should be included in future research.

Screening for STIs other than HPV could be very good indicator for monitoring women with HPV infection to prevent progression to cancer. We consider that this work represents our contribution to new knowledge about the prevalence and distribution of HPV genotypes in Croatia and the importance of screening, especially in the context that Croatia still does not have a nation-wide screening program for cervical cancer prevention, although this is being prepared.

## 5. Conclusions

The data from this population-based study on the prevalence and distribution of HPV genotypes in general female population of Southern Croatia (SDC), who underwent routine gynecological visits, will be valuable for better organization of HPV-based cervical cancer screening and vaccination program in this region and in Croatia. In this study, we found that the presence of some bacteria (*Ureaplasma*, *Chlamydia*, and *Gardnerella*) in the lower part of the female genital system affect the frequency of HPV infection and, consequently, can affect the faster progression to cervical dysplasia. We consider that including screening for sexually transmitted infections as monitoring in women with HPV infection could help identify women at risk for progression to cervical cancer.

## Figures and Tables

**Figure 1 viruses-15-00443-f001:**
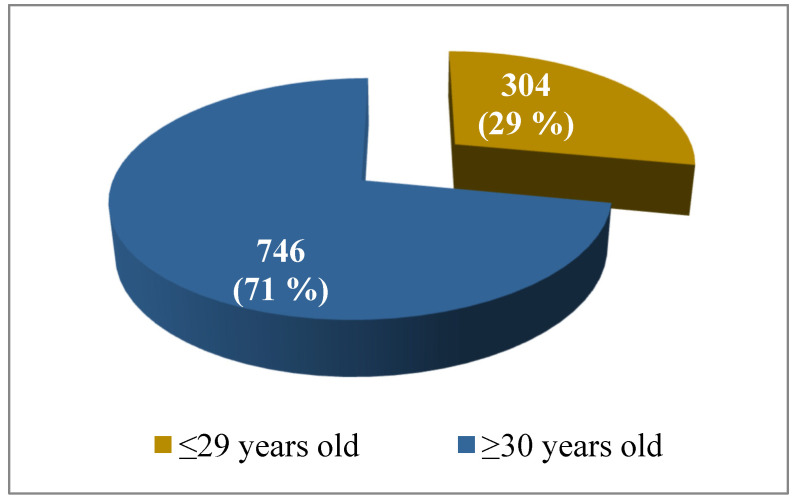
Age distribution in the total number of tested women (*N* = 1050).

**Figure 2 viruses-15-00443-f002:**
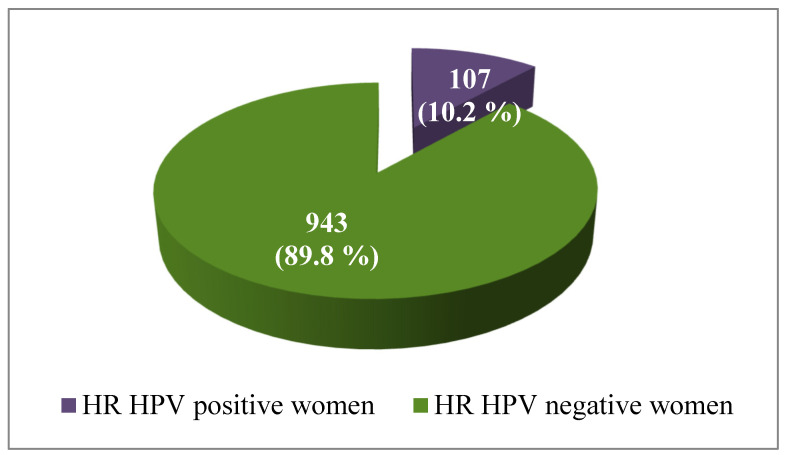
Frequency of HR-HPV positive in the total number of tested women (*N* = 1050).

**Table 1 viruses-15-00443-t001:** Demographic information of the study population (*N* = 1050).

Characteristics		*N*	Percentage (%)
**Civil Status**	*Single*	245	23%
	*Married*	805	77%
	**Total**	1050	100%
**Number of pregnancies**	*Nulliparous women*	314	30%
	*≤2 childbirths*	472	45%
	*≥3 childbirths*	264	25%
	**Total**	1050	100%
**Number of sexual partners in the last 6 months**	*No partner* *	78	7.4%
	*Established partner*	935	89.1%
	*≤2 new partners*	32	3.0%
	*≥3 new partners*	5	0.5%
	**Total**	1050	100%
**Use of contraceptives**	*Yes*	279	28.7%
	*No*	693	71.3%
	**Total**	972 **	100

* Without a partner at the time of the study (*N* = 78). ** Women who had a sexual partner at the time of the study (*N* = 972).

**Table 2 viruses-15-00443-t002:** Distribution of women with regard to microbiological results (*N* = 1050).

Microbiological Result	Positive	Negative	Total
** *Ureaplasma urealyticum* **	317 (30.2%)	733 (69.8%)	1050 (100%)
**HR-HPV**	107 (10.2%)	943 (89.8%)	1050 (100%)
** *Chlamydia trachomatis* **	21 (2.0%)	1029 (98.0%)	1050 (100%)
**CVM dysbiosis ***	181 (17.2%)	869 (82.8%)	1050 (100%)

* CVM dysbiosis (imbalance), some microorganisms present in pure culture or in the predominant number (*Streptococcus agalactiae*, *Escherichia coli*, *Gardnerella vaginalis*).

**Table 3 viruses-15-00443-t003:** Distribution of women according to cytological findings—Pap smear (*N* = 1050).

Pap smear	Number of women	Percentage (%)
**Normal**	1010	96.2
**LSIL**	37	3.5
**HSIL**	3	0.3
**Total**	1050	100.0

LSIL: low-grade intraepithelial lesion; HSIL: high-grade intraepithelial lesion.

**Table 4 viruses-15-00443-t004:** Frequency of HR-HPV positive women according to age (*N* = 1050).

Age	HR-HPV, *N* (%)		Total
	*Negative*	*Positive*	
**≤29 years old**	245 (80.6%)	59 (19.4%)	304 (100%)
**≥30 years old**	698 (93.6%)	48 (6.4%)	746 (100%)
**Total**	943 (89.8%)	107 (10.2%)	1050 (100%)

χ^2^ = 39.7215, *p* < 0.001.

**Table 5 viruses-15-00443-t005:** Correlation of a positive bacteriological result (Uu, CT, and GV) with a HR-HPV positive test result (*N* = 1050).

		HR-HPV, *N* (%)		Total
		*Negative*	*Positive*	
** *Ureaplasma urealyticum* **	** *Negative* **	687 (93.7%)	46 (6.3%)	733 (100%)
	** *Positive* **	256 (80.8%)	61 (19.2%)	317 (100%)
	** *Total* **	943 (89.8%)	107 (10.2%)	1050 (100%)
χ^2^ = 40.6591; *p* < 0.001				
** *Chlamydia trachomatis* **	** *Negative* **	929 (90.3%)	100 (9.7%)	1029 (100%)
	** *Positive* **	14 (66.7%)	7 (33.3%)	21 (100%)
	** *Total* **	943 (89.8%)	107 (10.2%)	1050 (100%)
χ^2^ = 12.5404; *p* < 0.001				
** *Gardnerella vaginalis* **	** *Negative* **	870 (90.6%)	90 (9.4%)	960 (100%)
	** *Positive* **	73 (81.1%)	17 (18.9%)	90 (100%)
	** *Total* **	943 (89.8%)	107 (10.2%)	1.050 (100%)
χ^2^ = 8.1381; *p* < 0.05				

**Table 6 viruses-15-00443-t006:** Correlation of a positive result for cervico-vaginal microbiota (CVM) dysbiosis (imbalance) with HR-HPV positive test result (*N* = 1050).

CVM Dysbiosis	HPV, *N* (%)		Total
	*Negative*	*Positive*	
** *Negative* **	785 (90.3%)	84 (9.7%)	869 (100%)
** *Positive* **	158 (87.3%)	23 (12.7%)	181 (100%)
**Total**	943 (89.8%)	107 (10.2%)	1050 (100%)

χ^2^ = 1.5135, *p* = 0.218598.

**Table 7 viruses-15-00443-t007:** Correlation of HR-HPV positive test result with cytological result (*N* = 1050).

HR-HPV Test Result	Pap Smear		Total
	*Normal*	*Abnormal*	
** *Negative* **	919 (97.5%)	24 (2.5%)	943 (100%)
** *Positive* **	91 (85.0%)	16 (15.0%)	107 (100%)
**Total**	1010 (96.2%)	40 (3.8%)	1050 (100%)

χ^2^ = 40.3758, *p* < 0.001.

**Table 8 viruses-15-00443-t008:** HPV type-specific prevalence among women in SDC (*N* = 1050) and genotypes distribution in HR-HPV positive women (*N* = 107).

	HR-HPV Genotypes Distribution		
	*Positive*	General Female Population (SDC) *N* = 1050	HR-HPV Positive Women *N* = 107
**HR-HPV genotype**	*N* = 134 **	Genotypes percentage	Genotypes percentage
**16**	38	3.6%	35.5%
**18**	7	0.7%	6.5%
**Other HR-types ***	89	8.5%	83.0%

* Other HR-HPV genotypes: 31, 33, 35, 39, 45, 51, 52, 56, 58, 59, 66, 68. ** Multiple infections were detected in 27 tested samples.

## Data Availability

Data presented in this study are available on request from the corresponding author.

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
