# Peer review of "Prevalence of Human Papillomavirus (HPV) among Females in the General Population of the Split and Dalmatia County and Its Association with Genital Microbiota and Infections: A Prospective Study"

_viruses, 2023, doi:10.3390/v15020443_

Round 1

Reviewer 1 Report

 This manuscript described the prevalence of HPV in general female populations of Southern Croatia and its association with general microbiota and infections. This manuscript is well organized. However, I would suggest making several revisions as shown below.

1.       Lines 54-57, Does this sentence mention the number of cases and death of cervical cancer?

2.       Lines 66-67, This study investigated partial HPV genotyping. Therefore, it should be changed to the distribution of HPV 16, 18, and other types.

3.       In Table 1, ASC-US and ASC-H do not be listed. Are there no cases of them?

4.       In Table 7, HPV positive rate seems to be low in the abnormal Pap smear group. Are there any reasons? For example, spontaneous elimination of HPV can be considered if the interval between Pap smear and HPV testing is long.

Author Response

Dear Reviewer 1

At the beginning, we would like to thank you for reading our manuscript and providing comments for it. The answers follow.

The responses to Reviewer 1 Comments

Point 1: Lines 54-57, Does this sentence mention the number of cases and death of cervical cancer?

Response 1: Yes, highlighted numbers refer to cervical cancer cases. Corrected.

Point 2: Lines 66-67, This study investigated partial HPV genotyping. Therefore, it should be changed to the distribution of HPV 16, 18, and other types.

Response 2: Corrected.

Point 3: In Table 1, ASC-US and ASC-H do not be listed. Are there no cases of them?

Response 3: In this study, there were no cases of ASC-US and ASC-H.

Point 4: In Table 7, HPV positive rate seems to be low in the abnormal Pap smear group. Are there any reasons? For example, spontaneous elimination of HPV can be considered if the interval between Pap smear and HPV testing is long.

Response 4: In this study, Pap smear and HPV test samples were taken simultaneously from women during regular gynecological examination in one visit. A large proportion of women who were HPV positive had a normal Pap smear because they probably had a transient HPV infection at the time of testing (at the time the study was conducted).

After the end of this study, we no longer monitored the women. However, it is still possible that in the further follow-up of the patients from this study, some of them might eventually develop premalignant lesions.

Reviewer 2 Report

Vanja et al. conduct a study in South Croatia to determine the prevalence of HPV, and the types of co-existing bacteria in cervical swabs. They find no association between HPV and dysbiosis of the cervical microbiome, but they do find correlation between certain bacteria and high-risk HPV types.

The study number is large and the data solid, so I have few points, mostly as suggestions for the discussion:

Other bacteria associated with bacterial vaginosis and with HPV include Sneathia and Provotella - Have the authors considered looking at these types?

The age groupings seem arbitrary - is there any reason for that split? e.g. did the under 29s have to access to prophylactic HPV vaccination?

Some minor sentence changes: Line 93  - the meaning of "were condition of" here is unclear, do the authors mean that normal CVM was defined by lactobacillus as dominant? Line 333 - "can influence to better receiving" is unclear, do the authors mean that dysbiosis predisposes patients to HPV infection, or promotes HPV infection? If the latter, please discuss or suggest mechanistically how this is the case. 

Author Response

Dear Reviewer 2

At the beginning, we would like to thank you for reading our manuscript and providing comments for it. The answers follow.

The responses to Reviewer 2 Comments

Point 1: Other bacteria associated with bacterial vaginosis and with HPV include Sneathia and Provotella - Have the authors considered looking at these types?

Response 1: We didn't detect bacteria Snaethia and Prevotella. Between bacteria associated with BV, we used detection of Gardnerella vaginalis as bacterial species with the highest correlation to progression among HPV positive women.

Point 2: The age groupings seem arbitrary - is there any reason for that split? e.g. did the under 29s have to access to prophylactic HPV vaccination?

Response 2: We divided women into two groups by age: ≤ 29 years old and ≥ 30 years old, because it is well known that the age limit of 30 years is important when talking about HPV infection and the possibility of developing cervical cancer. HPV infection is common in age group younger than 30, but these infections are transient and usually go away on their own. Cervical cancer is rare before the age of 30. However, for older than 30 years, the risk of persistent HPV infection and cervical precancer and cancer development is significantly higher.

Point 3: Some minor sentence changes: Line 93 - the meaning of "were condition of" here is unclear, do the authors mean that normal CVM was defined by lactobacillus as dominant?

Response 3: Corrected. The authors mean that normal CVM was defined by lactobacillus as dominant.

Point 4: Line 333 - "can influence to better receiving" is unclear, do the authors mean that dysbiosis predisposes patients to HPV infection, or promotes HPV infection? If the latter, please discuss or suggest mechanistically how this is the case. 

Response 4: Corrected. BV causes local immune system suppression, which can lead to greater susceptibility to HPV and, consequently, promote cervical dysplasia.